# LED Junction Temperature Measurement: From Steady State to Transient State

**DOI:** 10.3390/s24102974

**Published:** 2024-05-08

**Authors:** Xinyu Zhao, Honglin Gong, Lihong Zhu, Zhenyao Zheng, Yijun Lu

**Affiliations:** Department of Electronic Science, Xiamen University, Xiamen 361005, China; zxy1512753@163.com (X.Z.); gongh@xmu.edu.cn (H.G.); lhzhu@xmu.edu.cn (L.Z.); zzy@xmu.edu.cn (Z.Z.)

**Keywords:** high-speed signal, junction temperature, signal detection, LED, thermal reflectivity

## Abstract

In this review, we meticulously analyze and consolidate various techniques used for measuring the junction temperature of light-emitting diodes (LEDs) by examining recent advancements in the field as reported in the literature. We initiate our exploration by delineating the evolution of LED technology and underscore the criticality of junction temperature detection. Subsequently, we delve into two key facets of LED junction temperature assessment: steady-state and transient measurements. Beginning with an examination of innovations in steady-state junction temperature detection, we cover a spectrum of approaches ranging from traditional one-dimensional methods to more advanced three-dimensional techniques. These include micro-thermocouple, liquid crystal thermography (LCT), temperature sensitive optical parameters (TSOPs), and infrared (IR) thermography methods. We provide a comprehensive summary of the contributions made by researchers in this domain, while also elucidating the merits and demerits of each method. Transitioning to transient detection, we offer a detailed overview of various techniques such as the improved T3ster method, an enhanced one-dimensional continuous rectangular wave method (CRWM), and thermal reflection imaging. Additionally, we introduce novel methods leveraging high-speed camera technology and reflected light intensity (h-SCRLI), as well as micro high-speed transient imaging based on reflected light (μ_HSTI). Finally, we provide a critical appraisal of the advantages and limitations inherent in several transient detection methods and offer prognostications on future developments in this burgeoning field.

## 1. Introduction

The production of light through temperature rise has been an integral part of human history, dating back hundreds of thousands of years, when early modern humans utilized fire for heating and lighting purposes [1,2]. Flames, reaching temperatures close to 1000 °C (1273.5 K), were used to create torches and bonfires [3]. However, it was not until the 19th century that the rapid development of electricity laid the groundwork for modern lighting [4,5]. Despite these advancements, controlling temperature rise remains a significant challenge in light production. It took 40 years to address the issue of the oxidation of 3000 K filaments, starting with the demonstration of the constant electric lamp [6]. A significant breakthrough occurred in 1986 with the introduction of nitride-based blue-light-emitting diodes (LEDs), revolutionizing both people’s lives and the lighting industry. This innovation paved the way for efficient solid-state lighting [7]. Moreover, compared to traditional tungsten bulbs and fluorescent lamps, LEDs offer superior luminous efficiency and longer working lifetimes [8]. Additionally, with their advanced color control and modulation capabilities, LEDs have demonstrated immense potential across a variety of fields, including displays, imaging systems, communications, and transportation [9,10].

LEDs are renowned for their compact size, high brightness and exceptional luminous efficiency, rapid response time, high reliability, and eco-friendliness. They find diverse applications in fields such as high-resolution displays and sensing [11]. LEDs now have the capability to emit photons across all visible wavelengths. The AlGaAs material system is utilized for infrared and red emission; AlGaInP is used for amber, orange, and yellow-green; and AlInGaN is used for green to near-ultraviolet wavelengths. Group III nitride materials are ideal for achieving higher power LEDs, owing to their high thermal conductivity, electron saturation drift velocity, critical breakdown voltage, and fracture toughness against defect growth [12,13]. However, maintaining LEDs at low operating temperatures poses a challenge, considering their input power ratio compared to previous lighting technologies, and the fact that they can achieve the highest power conversion efficiency among all known artificial light sources. Thus, the PN junction temperature of an LED is a critical parameter for evaluating its performance, determining its service life, and ensuring its stability [14,15,16]. An increase in LED junction temperature can adversely affect luminous efficiency, underscoring the importance of its testing for semiconductor devices [17,18]. Preserving LED performance necessitates maintaining the junction temperature at a reasonable level. In LED-related research, the aging and degradation of LEDs resulting from temperature elevation have been prominent topics [19,20,21,22]. The temperatures of the junction and the phosphor layer (in phosphor-converted white LEDs) are the primary thermal issues for the next generation of high-power LED devices [23,24,25]. Precisely controlling LED temperatures is key to addressing these heat-induced issues [26,27]. Therefore, to provide suitable thermal management solutions, interpret photometric characteristics and packaging capabilities, and perform lifetime predictions, it is paramount that researchers can accurately determine both the steady-state and transient temperatures of LEDs across a wide range of conditions.

Over the past 20 years, numerous techniques for measuring junction temperature (T_j_) have been introduced by researchers, leading to significant advancements in many areas. However, accurately determining T_j_ in LEDs remains a prominent research focus, with researchers continually proposing new methods or refining existing ones. Among the existing measurement methods, those based on the optical and electrical properties of LEDs or involving physical contact predominate. However, this task is far from straightforward, as the majority of LED modules, including surface cover lenses and packaging components, pose challenges for temperature measurement. This challenge is further amplified when measuring transient temperatures in LEDs. Therefore, it is crucial to select appropriate technology to mitigate the limitations imposed by equipment requirements and operating conditions. There have been several reviews covering general temperature measurement techniques [28], semiconductor temperature measurement [29,30], and various electronic modules [31,32,33,34]. Numerous methods have been proposed for detecting the thermal characteristics of LEDs, with existing junction temperature measurement methods primarily focused on identifying temperature-sensitive parameters (TSPs), including temperature-sensitive electrical parameters (TSEPs) and temperature-sensitive optical parameters (TSOPs) [35,36,37,38,39]. Various LED junction temperature detection methods are prevalent in mainstream research. These methods encompass one-dimensional to two-dimensional (2D) distribution, transient LED junction temperature detection, and LED thermal imaging distribution. Common measurement techniques include the contact thermocouple method, the liquid crystal thermography (LCT) method, the non-contact infrared-sensitive camera test method, the heat reflection microscope test method, and other LED junction temperature detection approaches [40,41,42].

In this review, we delve into two key facets of LED junction temperature detection: steady-state and transient measurements. We introduce enhancements and advancements in existing methodologies, while also presenting the latest steady-state and transient LED junction temperature measurement technologies, considering both contact and non-contact methods. Within this scope, non-contact measurement methods are considered more suitable for the two-dimensional heat distribution measurement of LEDs, mitigating the risk of probe contact damage to the chip. Issues such as the spatial and temporal resolution, sensitivity, and precision of temperature are taken into account when analyzing and comparing various methods. This article outlines the advantages and drawbacks of different technologies and explores potential avenues for future development. The structure of the article is roughly as shown in Figure 1.

## 2. Steady-State LED Junction Temperature Detection Techniques

### 2.1. Micro-Thermocouple Method

Thermocouple thermometry, a prevalent method for contact temperature measurement, relies on the Seebeck effect, an inherent thermoelectric phenomenon. Put simply, a temperature differential at the two ends of a thermocouple generates an electromotive force, allowing assessment of the temperature at an unknown point by comparing the voltage difference between the measurement and reference connection points [43].

The micro-thermocouple method is a straightforward contact testing approach that utilizes commercially available thermocouples for device characterization. This approach entails the utilization of two distinct metals to create a thermocouple. When a temperature disparity arises between these metals, the inherent Seebeck voltage difference can be quantified. The advantages of thermocouple measurement are its cost-effectiveness and accuracy. However, it has limitations, such as a large volume, an incapability for thermal imaging, and the crucial consideration of thermocouple lung thermal mass, especially concerning the size of the device to be measured [44].

Thermocouples also face limitations in directly measuring the junction temperature (T_j_) of LEDs as they cannot access the active region. As an alternative, researchers must identify a proximal point, such as the surface, a solder joint, or an electrode, to gauge the temperature. They then evaluate the temperature based on the thermal resistance from that point to the junction [45,46]. For instance, Song et al. [47] determined the T_j_ of LEDs affixed to a thermoelectric cooler by employing solder joint temperature measurement techniques. They mounted a type-T thermocouple directly on the surface of the solder joint, assuming the junction-to-solder thermal resistance (R_j-sp_) of the LED to be 8 °C/W (8 K/W), as provided in the manufacturer’s datasheet. As a result, T_s_ varied between 65 and approximately 125 °C (338.15~398.15 K), with a constant interval of 15 °C (288.15 K), and within an input current range of 300 mA to 1000 mA, with a constant interval of 100 mA. In another study, Faranda et al. [48] evaluated the heat dissipation performance of a manufactured LED prototype by analyzing the reduction in the chip-on-board (COB) white LED’s T_j_. They measured the thermal resistance from the selected measuring point to the junction as 6.5 °C/W (6.5 K/W) using a FLUKE 54II thermometer and a temperature sensor. T_j_ varied between 50 and 56.5 °C (323.15~329.65 K) and 110.7 and approximately 123.2 °C (383.85~396.35 K). Jung and Lee [49] employed a solder temperature measurement technique to assess the heat dissipation efficiency of LED headlamps and determined a junction-to-solder resistance (R_j-sp_) of 1.7 °C/W (1.7 K/W). They attached a T-type thermocouple to the solder joint of the LED chip, and recorded a solder joint temperature (T_s_) of 62.8 °C (335.95 K) at thermal equilibrium. From the solder joint temperature measurement analysis, the T_j_ was calculated to be 103.6 °C (376.75 K), which was 6.4 °C (6.4 K) lower than the FE analysis result. In a recent study by Rammohan et al. [50], the T_j_ of a high-power LED array was determined using solder joint temperature measurement techniques. A K-type thermocouple was linked to the solder joints of individual LEDs, and the collective R_j-sp_ of the LED array, comprising six high-power LEDs, was measured at 2.75 °C/W (2.75 K/W) under an ambient temperature of 31 °C (304.15 K). Under different input power and environmental conditions, the experimental T_j_ value was between 45 °C (318.15 K) and 85.5 °C (358.65 K). The authors concluded that the thermography temperature map was consistent with the reference numerical working results and solder joint temperature measurements.

In a study by Xiao et al. [51], the overestimation of LED temperature measured by micro-thermocouples was attributed to several factors. These considerations encompassed the direct placement of the thermocouple on the LED surface, potential light absorption by both the thermocouple and the LED surface, and the reflection of light back into the LED’s active region. Although the authors contended that the surface of a 200 μm diameter thermocouple experiences minimal influence from incident or reflected light, they did not address the potential effects of light blocking on the LEDs themselves. Similarly, Shih et al. [52] utilized micromechanical monolithic thermocouples with probe sizes of 78 μm and 118 μm to perform electrical and thermal measurements on micro LEDs. Mechanical tests confirmed that the probe tip could make accurate contact with the miniature LED electrodes with low contact force, thus effectively determining the thermal and electrical characteristics.

In a recent investigation, Choi et al. [53] developed a microscale resistance temperature sensor based on Pt using a lift-and-drop technique, integrating it into an SMD-LED package for measuring T_j_. This method of fixing the Pt sensor’s position reduces errors and offers straightforward and dependable thermal characterization compared to less stable micro-thermocouples. The study demonstrated close agreement between the T_j_ temperature of the SMD-LED obtained using the microsensor and the results from numerical and structural thermal analyses. However, pre-calibration of the resistance temperature coefficient of the Pt microsensor is essential to ensure accurate thermal characterization.

### 2.2. Liquid Crystal Thermography (LCT) Method

The LCT method entails the application of a thin layer of liquid crystal onto the sur-face of the device being tested [54]. According to Csendes et al. [55], various types of liquid crystals can be utilized. One approach relies on a nematic-isotropic phase transition, offering a spatial resolution of 2 μm and a thermal resolution of 0.1 K. Dark spots emerge from phase transitions and are observable under a polarizing microscope. Because the phase transition of a specific liquid crystal occurs at a singular temperature, temperature measurement must be relative. The stage temperature needs to be adjusted relative to the transition temperature before determining the surface temperature. To generate a series of isotherms, the experiment must be repeated multiple times at different stage temperatures, potentially limiting its effective temperature range. Nevertheless, recent advancements in thermochromic liquid crystals have enhanced the convenience of liquid crystal thermal imaging (LCT) methods. Thermochromic liquid crystals change color within a specific temperature range, typically from red to blue. Crystals are designated by the temperature at which they activate and the range in which they operate. For example, ‘R40C5W’ indicates that the crystal turns red at 40 °C (313.15 K) and has a 5 °C (5 K) operating range, thus reflecting blue at 45 °C (318.15 K). The minimum range of color variation from red to blue is 2 °C (2 K), with a claimed thermal resolution of 0.1 °C (0.1 K), while spatial resolution is constrained by visible light diffraction to approximately 1 μm. Steps to achieve device-specific thermal imaging include coating the sample surface with “thin and uniform” black paint to enhance the resolution of reflected light from the liquid crystals. Furthermore, it is essential to determine the temperature range beforehand to select the appropriate liquid crystal. A typical experimental setup for measuring LED junction temperature using liquid crystal imaging is illustrated in Figure 2.

Lee and Park [57] pioneered the measurement of visible LED temperatures using nematic crystal thermography. They applied liquid crystals with transition temperatures ranging from 302 to 380 K to an LED chip. To counteract the LED’s strong optical power, a red filter was employed. Additionally, a high-power 660 nm laser beam served as the illumination source, ensuring that it did not contribute to device heating, as the LED chip was transparent at this wavelength. To achieve precise spatial temperature measurements, the authors recommended using a transparent, high-power laser. Despite leveraging the black and bright appearance of liquid crystals for evaluating LED device temperature and achieving resolutions of 21 and 35 μm, the measurement was constrained by the liquid crystal transition temperature.

### 2.3. Infrared (IR) Thermography Method

Infrared thermography is a real-time imaging technique utilized for measuring temperature distribution [58]. This method employs Planck’s law to gauge the LED junction temperature and monitor the corresponding two-dimensional distribution maps using an infrared camera. Planck’s blackbody law is applied to determine the object’s temperature, assuming a blackbody hypothesis to ascertain its absolute temperature. Thermal imaging cameras find application in thermal detection for various types of LEDs, including chip-on-board (COB) [59], surface-mount devices (SMDs) [60], and flip chips [61]. Presently, commercial infrared imaging systems offer temperature sensitivity ranging from 0.1 to 1 K, and temporal accuracy of 100 μs [62]. While the infrared region of the spectrum extends to 100 μm, temperature measurements typically utilize the 0.7–20 μm range due to reduced sensitivity beyond 20 μm [28]. A typical experimental setup for a thermal imaging camera comprises a lens with a known working distance, focusing the thermal radiation onto the camera’s detector. Depending on the orientation of the LEDs and environmental conditions, an adjustable emissivity setting should be employed for calibration at each location. However, as the actual object is not a true blackbody, understanding the wavelength radiation coefficient of each surface of the device under test is necessary for obtaining a more accurate thermal distribution map [63].

In their forward voltage verification thermal characterization study, Cengiz et al. [64] discovered that an emissivity value of 0.9 was suitable for the accurate calibration of chip infrared imaging for the phosphorescent conversion (pc) of LEDs with molded lenses. Meanwhile, Cheng et al. [65] conducted an evaluation of the thermal characteristics of red–green–blue (RGB)–white LEDs, utilizing a thermocouple-calibrated infrared camera, forward voltage methods (FVMs), and the finite element (FE) method while the transparent optical lens still covered the LED chip. Through the thermocouple calibration, they determined the emissivity values of red, green, and blue LEDs to be 0.90–0.94, 0.90–0.93, and 0.92–0.94, respectively. Significant variations in emissivity may arise from the transparency of the molded lens, infrared emissions, and reflections from other components of the LED module. Consequently, the authors proposed a correction factor to enhance the accuracy of their infrared thermography measurements. Their analysis revealed a temperature difference of up to 30 °C (303.15 K) between the corrected infrared measurements and the FVM, indicating that the surface temperature and T_j_ were not equal. Moreover, Chernyakov et al. [66] conducted a study on heat distribution in high-power flip InGaN/GaN blue LEDs using infrared thermal radiation, as depicted in Figure 3. They utilized an infrared camera with a short wavelength of 2.5–3 μm to improve spatial resolution to 3 μm. Additionally, they considered changing the emissivity of each LED section as a preliminary calibration for temperature control, aiming to provide a temperature accuracy of less than 2 K for infrared measurements.

Infrared thermography measurements face two significant challenges: diffraction-limited spatial resolution and local temperature uncertainty [67]. While the limited spatial resolution due to diffraction is an inherent drawback, the uncertainty of local temperature can be mitigated by calibrating the radiation characteristics of the surface being detected. However, accurately determining emissivity can be problematic due to significant variations in the heterostructure and radiation properties of the materials in the package. Semiconductor layers, metal electrodes, interconnects, coatings, and bonding elements exhibit different degrees of transparency or reflectivity to infrared radiation, leading to erroneous interpretations of the collected radiation and interference in the sublayer emissivity values.

### 2.4. Temperature Sensitive Optical Parameters (TSOPs)

The steady-state junction temperature characteristic of LEDs, T_j_, can also be indirectly measured using the intrinsic optical properties of LEDs. The emission spectrum of semiconductor devices is affected by temperature changes due to the temperature dependence of the band gap [68]. This observation has spurred LED researchers to estimate T_j_ through spectral power distribution (SPD) properties, particularly focusing on parameters such as peak wavelength and spectral bandwidth, collectively known as temperature-sensitive optical parameters (TSOPs) [69]. Methods utilizing TSOP measurements are non-destructive and do not disrupt LED electrical performance [70,71,72,73,74]. For instance, in the case of alternating current (AC) LEDs, T_j_ measurements based on TSOPs have been effectively implemented without affecting their electrical properties [75,76]. It is well-established that SPDs exhibit a redshift and broadening with increasing T_j_ [42]. The redshift primarily occurs due to the band gap reduction. Studies by Wang et al. [77,78] on GaN-based blue LEDs at low temperatures, and similar findings for high-brightness GaN on sapphire blue LEDs at elevated temperatures, underscore this phenomenon [79]. Additionally, it is noteworthy that SPDs demonstrate a blueshift and widening as the input current rises. This observation was corroborated by the research of Kim et al. [80], who investigated carrier leakage in GaN-based LEDs under forward bias conditions. An increase in current leakage was observed in the low series resistance of the LED, resulting in a blueshift of the SPD. The SPD also exhibited a blueshift and spread as the input current increased.

Furthermore, Li and colleagues [81] conducted a study exploring how variations in input current and temperature influenced the spectral characteristics of green InGaN/GaN multi-quantum well LEDs. Their investigation revealed that the excitation source had the capacity to alter the dynamics of carriers within the active region. Notably, they observed a pronounced blueshift at elevated input power levels, primarily attributed to the carrier shielding effect arising from the attenuation of the piezoelectric field. This phenomenon ultimately results in the quantum-confined Stark effect, as discussed in reference [82]. TSOP-based temperature measurement methods have been extended to 2D thermal imaging. Utilizing the new microscopic hyperspectral imaging (μ-HSI) technology, the two-dimensional spectral power distribution of the light-emitting surface can be obtained, and the temperature measurement of the LED surface can be combined with TSOP. Zhu et al. [83] utilized μ-HSI for two-dimensional junction temperature measurement of LEDs. This technique enables simultaneous capture of two-dimensional images and spectral information from the LED, allowing for the detection of junction temperature in both two-dimensional and three-dimensional (3D) spaces. By analyzing photons emitted from the side and epitaxial planes of flip RGB mini-LEDs, they derived junction temperature distributions in both two and three dimensions. Furthermore, the method provided insight into the ultrafine structure resolution between the sapphire substrate and epitaxial layer, and between the epitaxial layer and electrode. Jin et al. [84] investigated the 2D temperature distribution of blue, green, and red LEDs using a μ-HSI based centroid wavelength method (see Figure 4). After adjusting for the centroid wavelength coefficient, the authors were able to measure the surface temperature of the LED with a resolution as low as 3 μm, claiming to achieve sub-micron accuracy.

Lin et al. [85] applied this approach to investigate the impact of phosphor saturation on the two-dimensional temperature of fluorescence conversion LEDs. They introduced and defined the phosphor saturation factor (PSF), which increased with a rise in PC-LED driving current, indicating an increase in phosphor saturation. The authors analyzed and compared three representative feature points on the map. While the non-contact two-dimensional temperature distribution measurement method utilizing μ-HSI technology is suitable for the specified drive current range, it is important to note that, until relevant image processing methods are introduced in the future, there may still be some small errors that cannot be entirely eliminated using this method.

Lukas et al. [86] addressed temperature measurement challenges in parallel semiconductor devices caused by uneven temperature and current distribution due to device variations, leading to inaccurate LED temperature readings. They introduced a TSOP method that utilized the electroluminescence of SiC Mosfet to independently measure the junction temperature of each parallel semiconductor. This method capitalized on the unique temperature and current dependencies of the two sub-peaks in the electroluminescence spectrum of SiC Mosfet to calculate the temperature of a single semiconductor device by analyzing the intensity ratio of the sub-peaks. Additionally, to overcome electromagnetic interference and challenges in temperature measurement in high-voltage applications, they leveraged the temperature-dependent electroluminescence spectrum of Si IGBT and proposed a TSOP method. This method offered wide temperature and current detection ranges, and provided more sensitive detection at high current and low temperature [87].

### 2.5. Heterodyne Method

Operating similarly to the advanced stroboscopic oscilloscope, the heterodyne method utilizes sinusoidal/pulse signals for both exciting the device under test (DUT) and LED lighting. This induces natural mixing in the optical domain at the frequency difference between the temperature of DUT and the LED pulse. The left band of the reflected signal, captured with the charge-coupled device (CCD) camera, represents a slowly flickering mode. This technology significantly broadens the band width to tens of megahertz. However, it also imposes strict time constraints. Grauby applied the heterodyne method in heat reflection experiments by illuminating with a pulse frequency of 2F + f to capture temperature oscillations at 2F frequency. Consequently, the reflected signal included a slow flicker term with a frequency of f. The CCD samples the image at a frame rate of 4f. The heterodyne technique has been demonstrated to achieve a temperature resolution of around 10 mK. Utilizing blue light allows the spatial resolution to be reduced to 250 nm. However, achieving this maximum temperature resolution requires a lengthy average time of approximately 10 h or more, making it impractical. This approach is commonly referred to as the ‘four-bucket technique’, signifying the four image containers corresponding to the quarter-period integral of the reflected signal. Nevertheless, due to the time taken to read the CCD sensor and transmit data to the external processing unit, 10% of the integral is disregarded, introducing a slight margin of error [88]. However, some scholars have designed an improved method called the cyclic phase method based on the characteristics of the heterodyne method. The fundamental concept involves utilizing an LED pulse frequency of 2F, precisely matching the frequency of the studied thermal field. The frame rate and other CCD settings are flexible, providing the opportunity to optimize optical quality and dynamic range [89]. The utilization of a freely rotating camera mitigates stringent timing constraints, simplifying the phase locking to two function generators. Experimental results are depicted in Figure 5.

Compared to the traditional ‘four-bucket’ technology, this method is simpler to implement and offers higher accuracy. With a maximum frequency of 500 kHz, it provides an alternative for measuring the LED junction temperature, offering advantages in the realm of frequency-domain measurement.

This section has provided a brief overview of several steady-state junction temperature detection methods for LEDs, along with their respective advantages and limitations, which are summarized in Table 1. Thermocouples are a straightforward option for temperature monitoring, offering rapid responsiveness, ease of maintenance, and low cost. However, their accuracy is contingent upon the thermal resistance between the measurement point and the connection point, and their spatial resolution and response time are constrained by probe size and thermal capacitance, respectively. Moreover, thermocouples may experience self-heating due to light absorption, leading to potential overestimation of junction temperature during operation. Liquid crystal thermal imaging, while capable of high spatial resolution, is limited to detecting local hot spots on the LED surface and cannot accurately determine actual junction temperature. Its complexity, coating requirements, and uncertainty regarding thermal effects make it a less commonly used technology for LED temperature measurement. Infrared thermography offers real-time imaging for temperature distribution measurement. While it can be applied to various LED types, its spatial resolution is typically low, making it challenging to determine local temperature accurately. Additionally, environmental factors can introduce errors and influence experimental results. The TSOP method indirectly measures LED junction temperature by exploiting the relationship between semiconductor emission spectrum and temperature change. Although widely used for accurate temperature measurement, it necessitates precise measurement of each LED’s luminescence spectrum and clear visibility of the LED surface.

## 3. Transient LED Junction Temperature Detection Techniques

### 3.1. Transient Thermal Tester Method (T3ster)

Brain et al. [90] employed a modified T3ster method to extend the time-resolved measurement of constant drive current, based on JESD51, to dynamically measure the junction temperature under pulsed current conditions. Figure 6 illustrates a dynamic junction temperature measurement system comprising a T3ster (inclusive of its power supply for measuring junction temperature), a PC serving as a T3ster controller (for data acquisition and synchronization), and a pulse generation unit (equipped with a PC-controlled relay) responsible for converting the power supply’s output current into a current pulse to drive the LED. The rise and fall times of the current are significantly shorter than those of the temperature, ensuring accurate temperature response measurement. Additionally, 3D simulation of the junction temperature is conducted using Flotherm (see Figure 7).

Brain et al. [91] also introduced a method based on T3ster to predict the pulse temperature increase in vertical-cavity surface-emitting laser (VCSEL) and simplified the measurement process. When the DC and pulse bias currents are within the range of 0.5 to 2A, the frequency ranges from 10 to 10 kHz, and the duty cycle varies from 5 to 95%, and the junction temperature and optical output of the VCSEL sample are measured using the T3ster combined with the LIV measurement system.

János Hegedüs et al. [92] utilized the T3ster method to conduct transient measurements of the high-current temperature sensitivity function of LEDs in their investigation of pulse width modulation (PWM) LED aging. However, due to the simplified simulation process and the relationship between LED efficiency and temperature, the measured value in the switching part is lower than the analog value.

In brief, this method enables the visual measurement of the overall junction temperature of LEDs and is much simpler and more convenient than other measurement techniques. Furthermore, it allows for accurate measurement of not only the transient junction temperature of LEDs driven by constant current but also the transient junction temperature during PWM. However, its measurement resolution is limited, capable of measuring the transient junction temperature only at the millisecond level, and it is somewhat less effective for measuring the transient junction temperature at the nanosecond level.

### 3.2. Continuous Rectangular-Wave Method (CRWM)

Figure 8 presents graphical representations of altered LED voltage response patterns, depicted by dashed and dotted lines, arising from varying circuit switching speeds along with charging and discharging effects. Due to the significant relationship between junction temperature and forward voltage, the inability to promptly detect transient voltage changes indicates a deficiency in capturing comprehensive transient LED thermal data [93]. Consequently, the conventional forward voltage method (FVM) tends to yield lower junction temperature readings. To mitigate the limitations of FVM, Ze-Hui Liu et al. proposed the continuous rectangular-wave method (CRWM) [39].

Improving the accuracy of temperature response signals from LEDs involves addressing the shortcomings of FVM. This entails increasing the circuit switching speed and minimizing the time delay in signal acquisition. To achieve this, a continuous rectangular wave, shown in Figure 8a, is employed to drive the LED sample, eliminating the need for the circuit switching process. Accurate measurement of LED junction temperature is facilitated by ensuring that the rising and falling times of the rectangular wave are sufficiently short.

It takes time for the junction temperature to stabilize during both the heating and cooling processes. Ideally, as shown in Figure 8b, the junction temperature remains constant the moment the low current I_L_ transitions to a high current I_H_, and vice versa. By keeping I_L_ as low as possible, the steady junction temperature at I_L_ can be considered as the heat sink temperature T_hs_. Consequently, the temperature difference ΔT between T_hs_ and the junction temperature T_j_ at I_H_ can be used to calculate T_j_.

To begin with, a brief pulse current is administered to mitigate the self-heating effect and calibrate the temperature coefficient across various heat sink temperatures. Following this, the frequency and duty cycle of the rectangular-wave current utilized for driving the LED are set at 0.0625 Hz and 62.5%, respectively, with I_H_ = 300 mA and I_L_ = 1 mA. These parameters are determined through experimental optimization to ensure sufficient time for the LED to reach steady-state thermal equilibrium during each cycle. Monitoring of the rectangular-wave current waveform by an oscilloscope is carried out to guarantee a rapid rising time (approximately 15–20 µs) and to preempt any overshoot phenomenon. Moreover, the oscilloscope measures the LED voltage waveform to confirm the attainment of steady states at I_H_ and I_L_. The heat sink temperature is incrementally raised from 25.0 (298.15 K) to 65.0 °C (338.15 K), and the transient voltage waveform of the LED is recorded at each corresponding heat sink temperature using the oscilloscope (refer to Figure 9 with T_hs_ = 25.0 °C (298.15 K)). Subsequently, the voltage waveform from the rising edge to the steady state of each cycle is isolated to capture ΔV_rise_ = V_max_ − V_SH_. Employing multi-cycle superposition can further enhance the signal-to-noise ratio when multiple periodic current signals are used for continuous LED sample driving. Furthermore, Figure 10 illustrates that the voltage signal at the falling edge appears to be more visibly disturbed by noise or discharging effects, resulting in a less favorable signal ΔV_fall_ compared to ΔV_rise_. Similar observations are noted across various samples under differing currents and heat sink temperatures. Consequently, ΔV_rise_ is deemed more suitable than ΔV_fall_ for determining ΔT.

The continuous rectangular-wave method (CRWM) was introduced as a solution to overcome the limitations associated with conventional FVM, including switch-induced acquisition delays and charging/discharging effects. CRWM operates in either voltage mode or current mode for junction temperature measurements of LEDs. CRWM achieves more precise determination of the junction temperature by employing a switch-free periodic rectangular wave to drive the LEDs and capturing the resulting heating curve with a high-definition, high-speed oscilloscope. Both experimental results and simulations have confirmed that rectangular waves with rising times of less than 40 µs effectively mitigate the charging effect.

### 3.3. Micro-Raman Spectroscopy

The development of Raman thermal imaging has effectively improved the spatial resolution encountered in infrared technology [94,95,96], enabling the detection of sub-micron regions in semiconductor devices. This section provides a concise overview of the evolution of time-resolved Raman thermal imaging for the sub-micron temperature measurement of semiconductor devices at two-dimensional and three-dimensional temperatures [97].

Raman spectroscopy provides a non-contact method for mapping temperatures, offering high spatial resolution down to 1 μm [98] and temporal resolution reaching up to 200 ns [99]. Utilizing this approach, the temperature distribution of the active regions of semiconductors, delineated by the Raman active material layer, can be assessed based on their phonon frequencies. A standard Raman setup includes an excitation source, typically a laser, directed onto an LED, a beam splitter, a sample holder, and a spectrometer capable of detecting frequency shifts, as depicted in Figure 10. Various lasers can be utilized as excitation sources, including argon ions (488.0 and 514.5 nm), krypton ions (530.9 and 647.1 nm), He:Ne (632.8 nm), Nd:YAG (1064 and 532 nm), and diode lasers (630 and 780 nm) [100]. To mitigate temperature measurement errors stemming from the excitation source and carrier generation, the excitation source can be lowered, or a source with a wavelength lower than the LED band gap can be selected. It is important to note that Raman spectroscopy temperature measurements can be time-consuming over large surface areas, primarily due to the necessity of raster scanning and data integration. Furthermore, acquiring fine temperature distributions can also be time-consuming, particularly when detecting weak Raman signals. Nevertheless, Raman spectroscopy techniques are adept at capturing temperature distributions of features at the micron scale with remarkable spatial resolution. Consequently, researchers frequently utilize them for thermal characterization endeavors.

**Figure 10 sensors-24-02974-f010:**
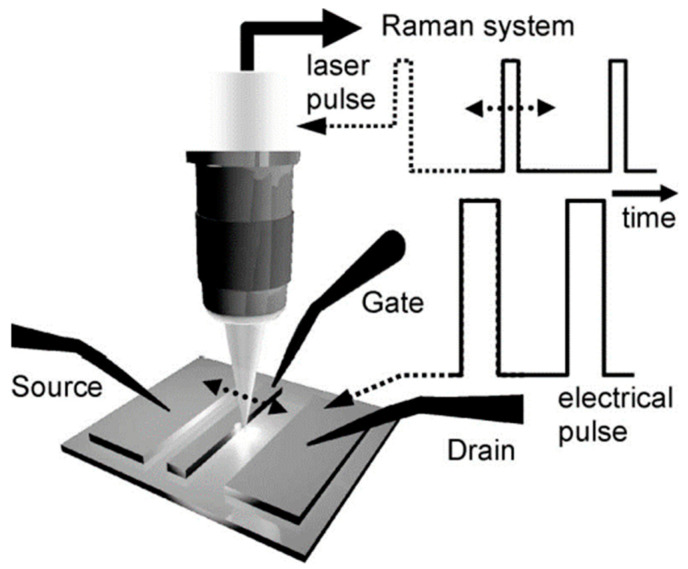
Schematic diagram of time-resolved micro-Raman thermal imaging experimental device [99].

Initially, Raman spectroscopy is employed to measure the device’s temperature based on the temperature-dependent phonon shift resulting from the device’s self-heating. The time evolution of the device temperature is captured using the variable laser pulse time delay associated with the electrical excitation of the device. In the future, nanosecond or picosecond pulsed laser sources will likely be used to detect operating frequencies of up to gigahertz. The equipment operates with square voltage pulses, resulting in a 5–10% reduction in current over the pulse duration. The XY stage scans the device under the focused laser beam with a step size of 0.2 μm to acquire spatial information. The lateral spatial resolution is determined by the laser spot size, estimated to be 0.5–0.7 μm at the contact edge through laser beam scanning. Figure 5 depicts the experimental setup. This configuration deviates from standard Raman thermal imaging, which typically uses a continuous-wave laser to determine the time-averaged temperature of a DC-operating or pulsed device.

Figure 11a,b display the temperature traces recorded at the center of non-gated AlGaN/GaN devices grown on SiC and sapphire substrates, respectively. Operating with 2 μs long electrical pulses and a 50% duty cycle, these devices exhibit adiabatic heating in the first few nanoseconds, followed by thermal diffusion. The initial turn-on phase shows a temperature rise below 200 ns, resulting from a combination of adiabatic heating and thermal diffusion. Similarly, a rapid initial temperature decay is observed when the device is turned off. However, within 2 μs, neither the Sic-based nor the sapphire-based device reaches a truly stable temperature.

Figure 12a,b illustrates the thermal diffusion in the SiC substrate for the non-gated AlGaN/GaN device center. The temperature is determined by focusing the probe laser beam on the device or at different depths of the SiC substrate. The drop in temperature towards SiC and the radiator is slower than the rises and falls in temperature rises in the device. This pattern is corroborated by the simulations shown in Figure 12a, where the difference is attributed to simulation parameter uncertainty. The simulation also highlights the time delay in substrate heating and cooling, which increases with depth.

In brief, the progress in time-resolved Raman thermal imaging technology has facilitated the two- and three-dimensional thermal analysis of semiconductor devices with sub-micron spatial and high temporal resolution. Initially, the temperature change occurs within a timeframe slower than 200 ns, followed by a gradual change in substrate temperature, which is slower and less abrupt compared to that in the device.

### 3.4. Thermal Reflection Imaging Method

Thermal reflection microscopy is a technique for high-resolution thermal profiling without physical contact. It operates by measuring the relative alterations in surface reflectance and utilizes this data to map out the temperature distribution accurately [101]. Although the temperature dependence of material reflectivity is typically between 10^−5^ and 10^−4^ K^−1^ [102], temperature-induced variations can be detected by combining sensitive (amplified) measurement systems. The advantage of thermally reflective imaging is its ultra-fast temporal resolution of up to 800 ps, or 200–250 nm if illuminated with ultraviolet or visible light [103]. In addition, thermal reflectivity can be used for the optimization of a variety of materials and has been used to measure the temperature of various electronic components.

The thermal reflection imaging method can measure small changes in visible light reflectivity related to temperature fluctuations. By generating thermal images, it can display the temperature curve of organic light-emitting diodes (OLEDs) over time, and the thermal crosstalk of adjacent pixels can be observed. The experimental setup is illustrated in Figure 13, utilizing an inorganic LED as the light source for the optical microscope, reflected from the OLED display, and imaged by the CCD camera [104].

Kendig et al. [105] used thermal reflection imaging to determine the two-dimensional temperature pattern of encapsulated UV and blue LEDs. A custom-built one million-pixel CCD system was used to acquire a heat-reflective signal and a phase-locked technique was used to improve measurement resolution. Wavelength, surface roughness, and material-dependent heat reflection coefficients were calibrated to measure quantitative temperature values. The findings demonstrated that thermal reflectometry can be used to study the thermal inhomogeneity and transient thermal response of LEDs.

Reliable reflectivity measurements encounter two significant obstacles. Firstly, in heat reflection measurements, a common approach involves using a simple locking technique, where the LED operates continuously, and the camera and the device under test are synchronized. However, the luminescence of the device’s pixels can be problematic, as the emitted light may overshadow the subtle changes in the reflectivity of the incident LED light. It is not sufficient to rely solely on optical filtering, as the wavelength probe LED has broad pixel spectral overlap. Secondly, considering that the sought variation in reflectivity is very small (<10^−3^), each camera exposure must capture nearly the same light intensity to minimize the impact of camera nonlinearity. To address these challenges, the relative timing of the camera, LED, and pixel operations should be meticulously selected, as demonstrated in the experiment depicted in Figure 14.

Katz et al. [104] quantified changes in reflectivity normalization concerning commercial OLED displays, albeit with a limitation in resolution. While a time resolution of approximately milliseconds can be achieved in the measurement of junction temperature, it is essential to acknowledge the low resolution inherent in this approach. The study delves into the thermal crosstalk between pixels, revealing that the operation of individual pixels can impact the lifespan of adjacent pixels. Furthermore, the elevated temperature resulting from this interaction accelerates the degradation of OLEDs.

Xiao et al. [51] devised a method to ascertain the junction temperature (T_j_) by scrutinizing the relative reflection intensity of the incident excitation light. They employed a phase-locked method to extract luminous light interference from LEDs, which enhanced the measurement’s dynamic range. Their methodology entailed calibration using micro-thermocouples. Nevertheless, they highlighted apprehensions regarding the utilization of thermocouples for calibration, citing issues such as variations in diameter between the excitation spot and the thermocouple, along with concerns about the uniformity of heat distribution on the chip surface.

### 3.5. High-Speed Camera and Reflected Light Intensity Method (h-SCRLI)

This approach leverages the linear temperature-dependent surface reflectivity principle of GaN LEDs. A high-speed camera captures reflected photons, from which a 2D temperature distribution is derived by analyzing changes in light intensity [106]. The relationship between the reflectivity of GaN LEDs and temperature can be expressed as follows [107]:T_s_ = T_0_ + K_L_ [L (T_s_) − L (T_0_)] = T_0_ + K_L_ΔL(1)

Here, T_s_ denotes the surface temperature of the GaN LED, while KL represents the temperature sensitive parameter (TSP). The variables L(T_s_) and L(T_0_) correspond to the reflected light intensities at temperatures T_s_ and T_0_, respectively, and ΔL indicates the relative change in the reflected light intensity.

Before calculating the two-dimensional transient temperature distribution of the LED under test (LUT), it is necessary to preprocess the two-dimensional reflected light intensity image obtained from the camera. This includes tasks such as rotating and aligning the image, as well as extracting the contour of the chip from the entire image. To achieve this, Lin et al. [107] utilize the sub-pixel Shi-Tomasi corner detection algorithm to identify the contour of the LED chip. The formula for determining the final response value of the R corner is as follows:R = min (λ_1_, λ_2_)(2)
where λ_1_ and λ_2_ represent the eigenvalues of the gradient matrix. The algorithm employs a local window h (m, n) that moves on the image, determining the position of significant grayscale changes, and corners are identified when the grayscale value of the window changes significantly in all directions. As the orientation of the LED chip may vary during experimentation, the bilinear difference and inverse mapping methods are employed to rotate and align the image, ensuring accurate and convenient extraction of the LED profile.

An experimental setup for measuring the dynamic two-dimensional temperature distribution of LEDs is shown in Figure 15a. Figure 15b shows the trend of the reflection intensity and LUT transient temperature acquired using a high-speed camera with a periodic drive current. The high-speed camera is utilized to capture the distribution of two-dimensional transient reflection intensity. Subsequently, employing the negative linear relationship between reflection intensity and temperature, the corresponding two-dimensional transient thermal characteristics can be derived, particularly during the heating and cooling processes.

Figure 16 showcases a series of transient two-dimensional temperature distributions using the h-SCRLI method during the rising edge of the driving current, along with a comparison with the transient response of the thermal reflection imaging method [17]. Figure 16 exhibits a sequence of transient 2D temperature distributions of the LUT utilizing the h-SCRLI method during the rising edge of the drive current. It also includes a comparison of transient responses between the h-SCRLI method and the thermal reflection imaging method. Notably, the electrodes stand out distinctly from other areas due to significant temperature discrepancies, indicating a relatively low thermal sensitive parameter (TSP) of the electrodes, and their insensitivity to reflective light. Although both methods show consistent temperature trends, the h-SCRLI method exhibits superior temporal resolution. The temperature distribution, with a resolution of 528 × 512 pixels, appears quite uniform across the chip’s surface. An average transient temperature distribution extracted from a 30 × 70 pixels box is compared with that obtained using the TI method. It demonstrates that a notably steeper curve is measured using the h-SCRLI method, indicating a faster time response. In Figure 17e, a similar trend is observed at the falling edge, where the curve measured using the h-SCRLI method decays more rapidly than that measured using the thermal reflection imaging method. Furthermore, an analysis of the curves obtained with the h-SCRLI method indicates that it takes approximately 140 ms to reach a steady state during the rising edge (heating process), while it takes about 80 ms to reach steady state during the falling edge (cooling process). This observation aligns with the corresponding curves obtained with the thermal reflection imaging method, confirming that the cooling process is indeed faster than the heating process.

This non-contact high-speed camera, which operates by detecting reflected light, is employed to detect the transient two-dimensional temperature of the LED chip. It has a finely detailed spatial resolution, down to the sub-micron scale, achievable with an optical microscope. At the highest frame rate of the high-speed camera, the time resolution can achieve 505 μs at 1980 fps, 179 μs at 5600 fps, and 68 μs at 14,600 fps. However, it is worth noting that a reduced exposure time at higher frame rates may introduce more noise and could potentially affect the uniformity of the two-dimensional temperature distribution.

### 3.6. Micro High-Speed Transient Imaging Based on Reflected Light (μ_HSTI)

This technique, termed micro high-speed transient imaging based on reflected light (µ_HSTI), investigates a method for analyzing two-dimensional transient junction temperature distributions. Particularly, when stimulated by a nanosecond periodic short-pulse signal, a standard camera captures 2D transient thermal reflection images, achieving temporal resolution on the nanosecond scale. Figure 18 illustrates the operation of driving the LUT using pulse signals during the heating phase and the sampling signal during the cooling phase. In Figure 18a, the LUT experiences excitation from various short pulse widths, each representing the average transient integration time. The LED chip is stimulated by nanosecond high-speed pulse signals. Due to the relatively weak pulse signal, the exposure time can be adjusted to sample multiple cycles repeatedly, thereby acquiring the time–response curve by sequentially altering the pulse width. The time resolution is contingent upon the minimum pulse width. To capture multi-cycle image signals, a longer CCD exposure time is configured, resulting in a substantial enhancement in the signal-to-noise ratio [100,101,102,103,104,105,106,107,108,109,110]. Figure 18b demonstrates the setting of the sampling pulse signal during the cooling process according to the Boxcar principle. A digital delay generator provides an external reference trigger to synchronize the camera. Signals corresponding to different stages of the cooling process are collected sequentially as the pulse delay shifts. This process continues until the entire cooling process is captured. The sampling pulse is synchronized with the camera’s exposure time, allowing the determination of the minimum sampling time based on the camera’s minimum exposure time. The experimental device’s schematic diagram is depicted in Figure 19.

As depicted in Figure 19, the schematic layout for capturing the transient two-dimensional temperature distribution of the LEDs comprises various components: an arbitrary function signal generator, a CCD camera, an optical microscope equipped with filters, an optical fiber, incident light, the LUT, a precision source/measure unit, and temperature control devices. The incident light is emitted by a high-power near-infrared chip, which integrates four 690 nm LEDs. The LUT utilized in this setup is a 408 nm blue bare LED with a chip area of 1 mm × 1 mm. To prevent additional excitation of photoluminescent emission from the LUT and band gap modulation, incident light is selected with a significantly longer wavelength than that of the LUT. Figure 20 and Figure 21 illustrate the transient two-dimensional junction temperature distribution of the LUT at various time points during both the heating and cooling phases. The resolutions for these temperature distributions are in the range of nanoseconds and microseconds, respectively. Figure 20a–d illustrate the two-dimensional transient junction temperature distribution of the LUT driven by different pulse widths during heating. Figure 20e represents the time–response curve of the average transient junction temperature in the 40 × 40-pixel region in the logarithmic coordinate, as indicated in the red box in the illustration.

By applying a periodic pulse signal to LUT, researchers calculated both the TSP and the intensity of reflected light within a single period. However, achieving a minimum resolution of 5 nanoseconds surpassed the capabilities of standard cameras in terms of time resolution. Initially, due to the extremely brief signal acquisition time, the variation in the two-dimensional temperature distribution was not readily discernible. As depicted in the inset of Figure 20e, the temperature rose by approximately 1.2 °C (1.2 K) from 5 ns to 50 ns. Subsequently, temperature elevation commenced at around 10 microseconds, peaking at approximately 10 ms. Figure 21 illustrates the time response of the cooling process. Similar to the heating phase, a 40 × 40 pixel region marked by a red square box was used to depict the average transient temperature curve, as shown in the inset of Figure 21e. The temporal resolution during the cooling process was in the microseconds, subject to the minimum exposure time of the CCD camera. In brief, Wang et al. successfully surmounted the time resolution limitations of standard cameras by utilizing a periodic short pulse signal to drive the LED chip. They were able to achieve a two-dimensional transient junction temperature distribution during the heating phase while preserving spatial accuracy, with a temporal resolution as fine as nanoseconds. The temporal resolution of the method was determined by the pulse width of the signal generator. During the cooling process, the two-dimensional transient cooling temperature distribution with a minimum exposure time of 1 μs was sequentially obtained by using the Boxcar gated integration technique.

This section has presented the main methods for detecting the transient junction temperature of LEDs introduced in this review. Furthermore, we compare the methodologies employed by various investigators and summarize the advantages and limitations of each approach in Table 2.

## 4. Summary

This review commences by tracing the evolution of LEDs, with a particular focus on thermal challenges that hinder their optimal performance. It underscores the critical importance of LED junction temperature (T_j_) detection in enhancing LED efficiency and extending operational lifespan. Given the intricate nature of addressing thermal concerns through cooling designs, photometric analysis, and packaging strategies, this review provides a comprehensive exploration of various methods for LED T_j_ detection, covering both steady-state and transient detection approaches. These methodologies are dissected to elucidate their underlying principles and applicability across different LED scenarios. Notably, the review conducts detailed comparisons of experimental parameters employed by different researchers, encompassing both steady-state and transient T_j_ measurements. In the transient detection domain, the review offers an in-depth examination of techniques such as micro-Raman spectroscopy, an improved one-dimensional continuous rectangular wave method, thermal reflection imaging, as well as novel methods utilizing high-speed camera and reflected light intensity, and micro high-speed transient imaging based on reflected light. Furthermore, alongside presenting research findings and discussions, the review thoroughly assesses the strengths and potential challenges associated with each measurement method, serving as valuable guidance for researchers engaged in LED junction temperature studies.

## Figures and Tables

**Figure 1 sensors-24-02974-f001:**
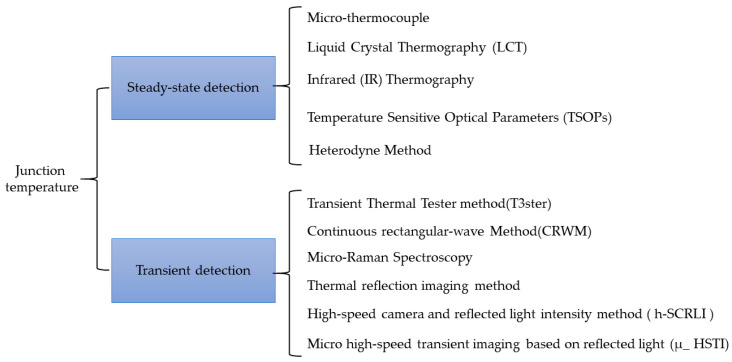
Schematic diagram of the structure of the article.

**Figure 2 sensors-24-02974-f002:**
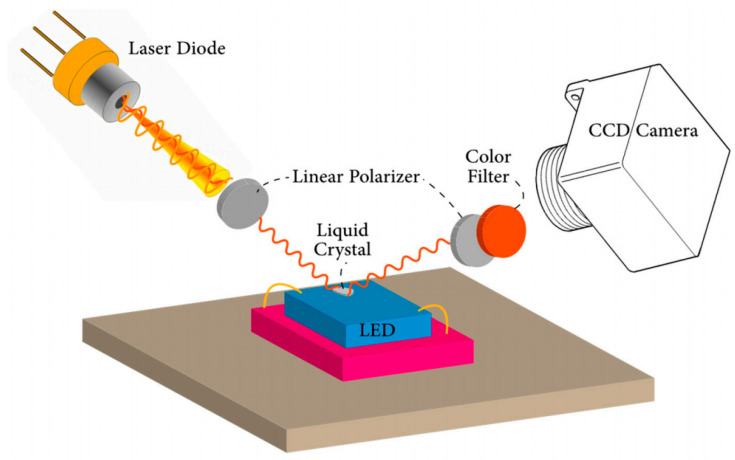
Schematic diagram of the experimental arrangement of liquid crystal thermal imaging technology. The experimental setup usually consists of a polarized laser beam, a charge-coupled camera with a color filter, and a liquid crystal covering the surface of the LED [56].

**Figure 3 sensors-24-02974-f003:**
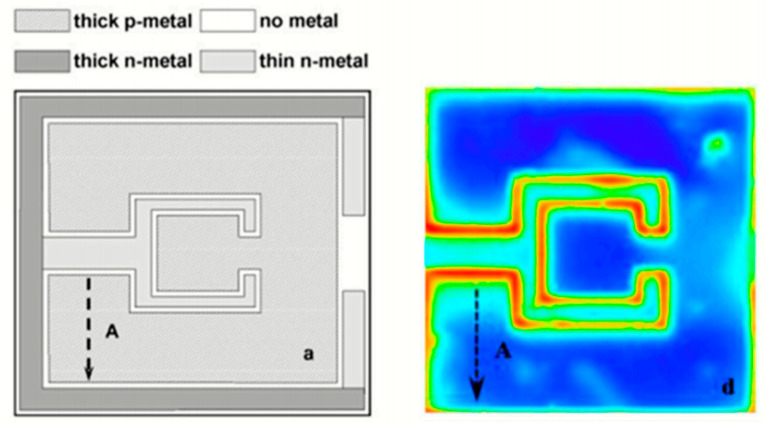
Lateral infrared distribution image of flip LEDs prepared by photolithography and dry reaction etching techniques. The schematic of the chip is shown on the left, while the captured infrared radiation is shown on the right [66].

**Figure 4 sensors-24-02974-f004:**
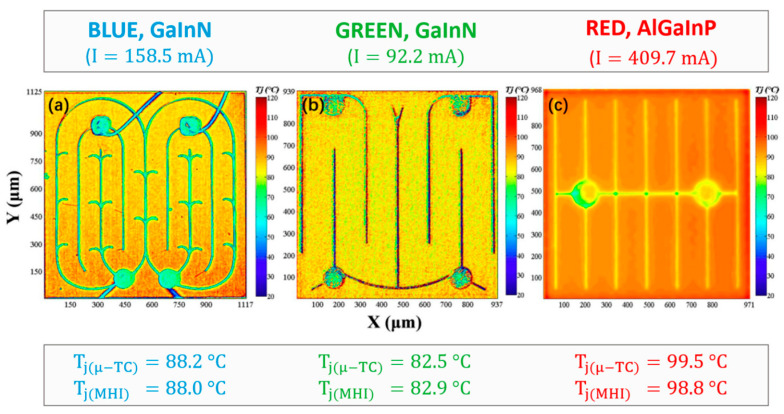
μ-HSI was utilized to measure the two-dimensional temperature distribution of blue (**a**), green (**b**), and red (**c**) LEDs at a heat sink temperature of 75 °C (348.15 K). Information regarding the color, material system, and driving current of each LED is provided at the top, while the junction temperature (T_j_) measured by both micro-thermocouple and μ-HSI is presented at the bottom. The average standard deviation of T_j_ (μ-TC) and T_j_ (μ-HSI) was recorded as 0.9 °C (0.9 K) [84].

**Figure 5 sensors-24-02974-f005:**
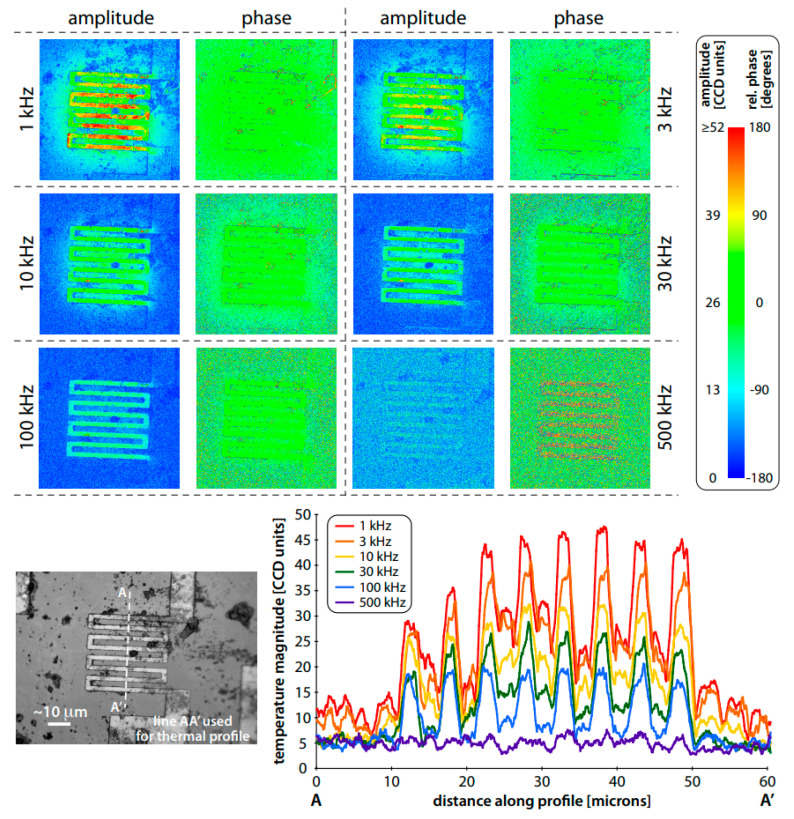
Frequency domain thermoreflectance imaging of a 40 μm heater with ‘cyclic phase lag’ heterodyne locking (the frequency shown is the thermal frequency) [89].

**Figure 6 sensors-24-02974-f006:**
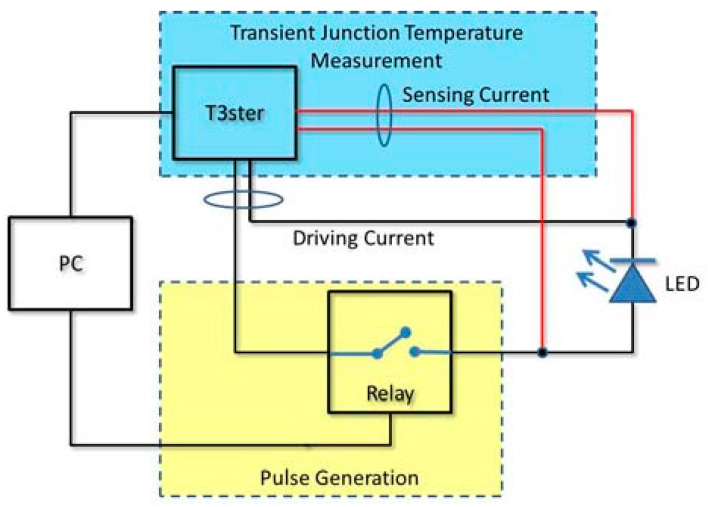
T3ster junction temperature measurement system.

**Figure 7 sensors-24-02974-f007:**
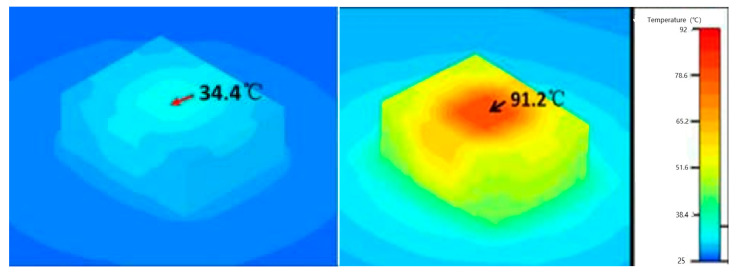
Simulation using Flotherm to drive dynamic junction temperature peaks of LEDs with pulse trains.

**Figure 8 sensors-24-02974-f008:**
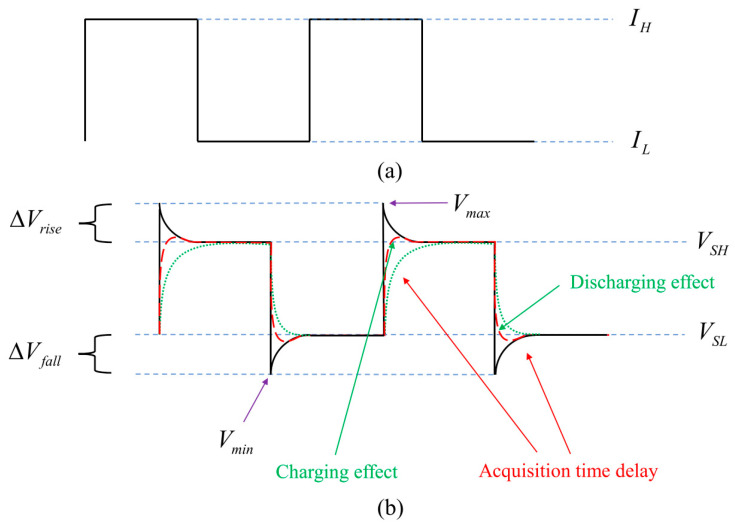
(**a**) Continuous rectangular-wave current. (**b**) Different voltage response waveforms of LED [39].

**Figure 9 sensors-24-02974-f009:**
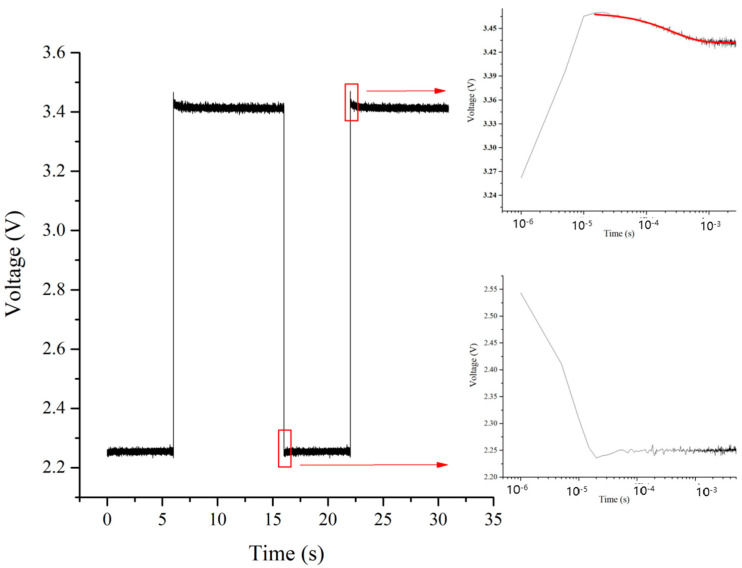
Voltage waveform of the LED driven by rectangular-wave current [94].

**Figure 11 sensors-24-02974-f011:**
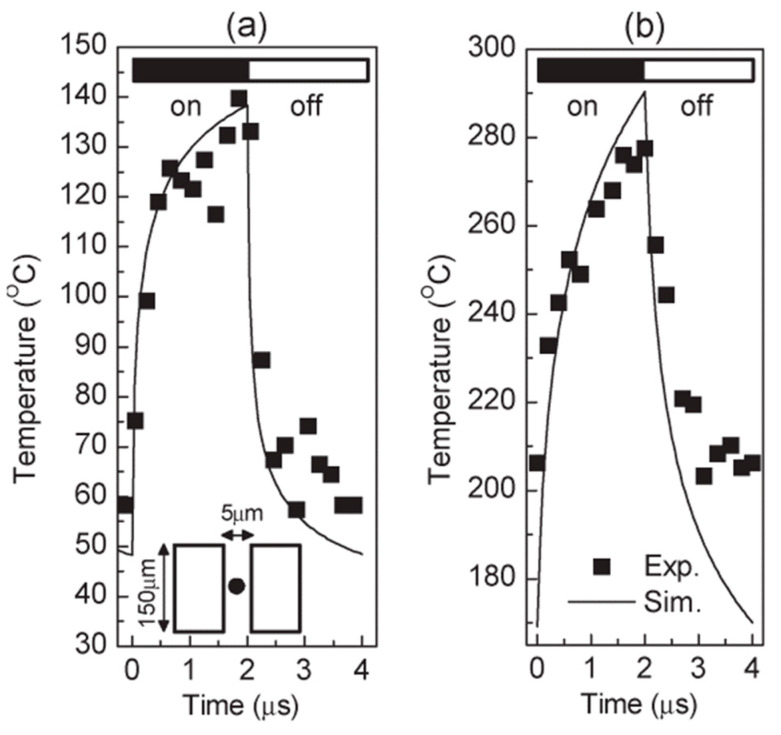
The variation of the center temperature of the mesa isolated AlGaN/GaN device grown on (**a**) SiC substrate and (**b**) sapphire substrate with time [99].

**Figure 12 sensors-24-02974-f012:**
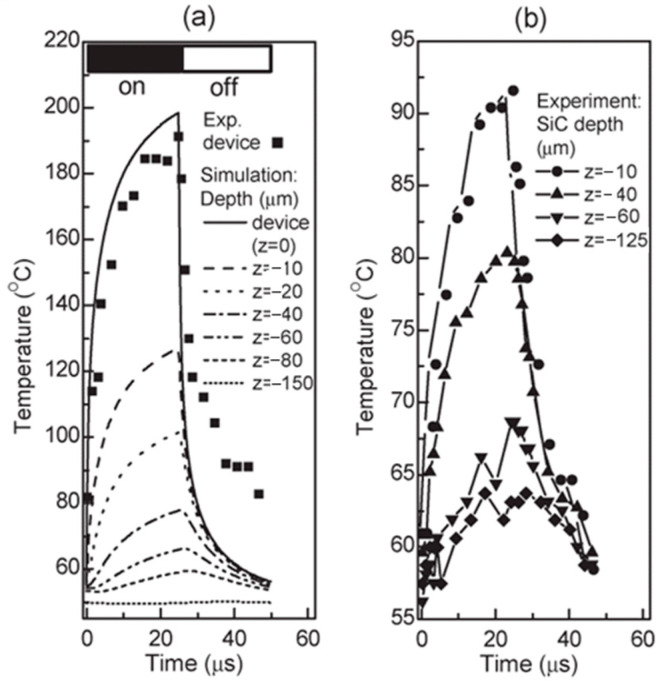
(**a**) The variation of the center temperature of a 20 μm wide ungated AlGaN/GaN device on SiC substrate with time, and the simulated temperature evolution at different depths (z) in the device and SiC substrate. (**b**) The measured temperature evolution at different depths in the SiC substrate. The device operates at 25 μs long 40 V (159 mA) square bias pulse and a 50% duty cycle [99].

**Figure 13 sensors-24-02974-f013:**
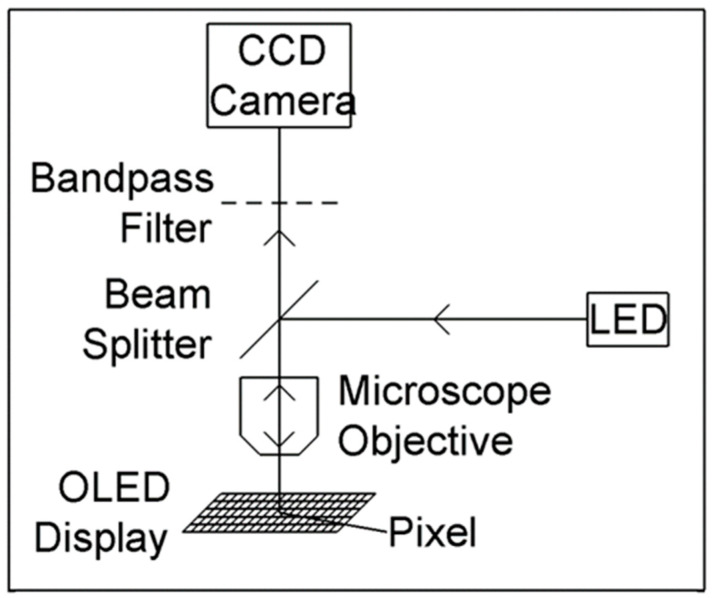
Schematic diagram of the heat reflection imaging experimental device [104].

**Figure 14 sensors-24-02974-f014:**
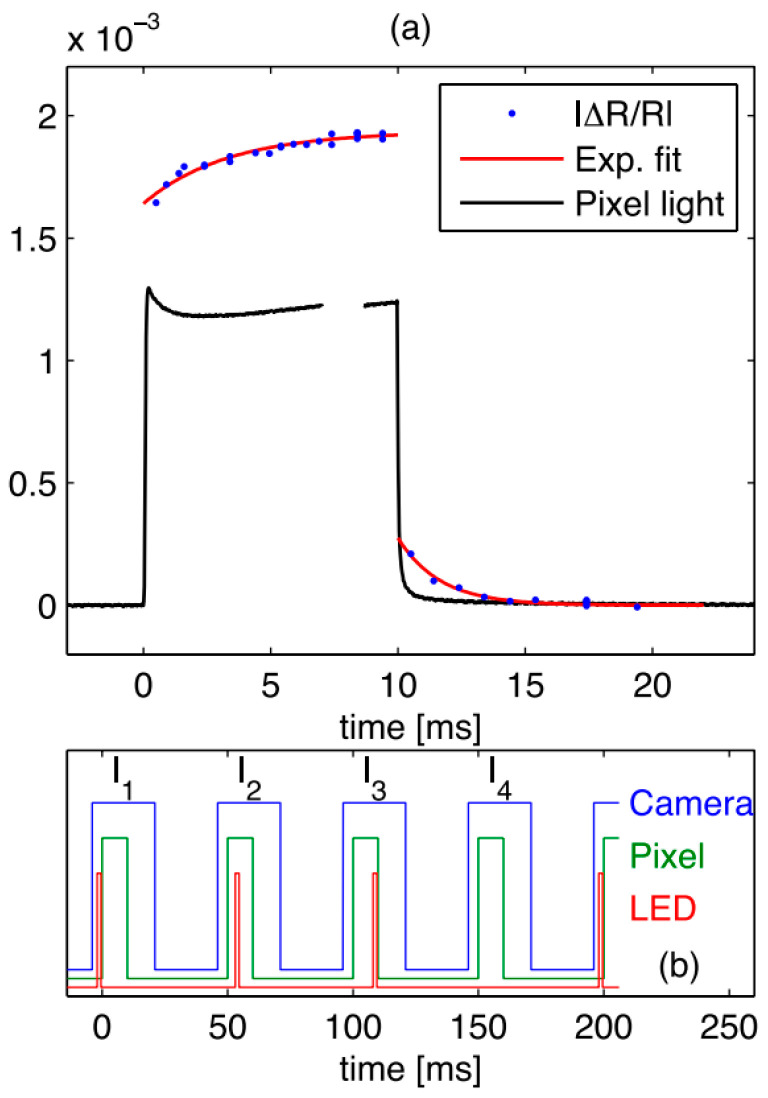
(**a**) |μ R/R| is a function of time, pixel offset to 12 V 10 ms; (**b**) Timing diagram: the timing of the LED pulse relative to the pixel pulse changes; four function generators are used to generate various pulses, with LED pulses of 1 ms [104].

**Figure 15 sensors-24-02974-f015:**
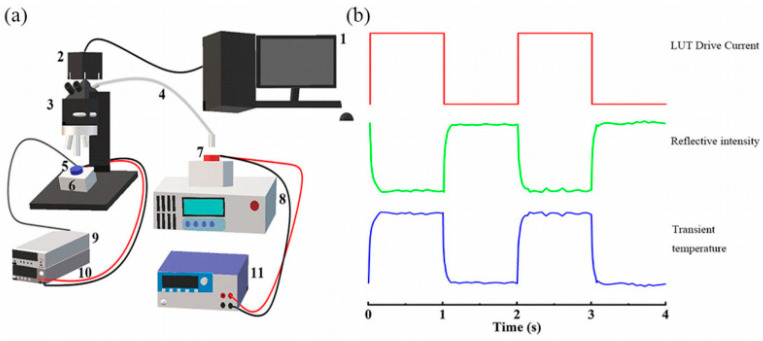
(**a**) The setup used in the experiment to measure the dynamic two-dimensional temperature distribution. The measurement setup mainly contains highspeed camera (2), microscope with a high-pass filter (3), blue LUT (5), incident red LED (7), Heat sink (6) with temperature controller (8) (9), electrical source meter (10) (11) for blue LUT and incident red LED. (**b**) An illustration detailing the drive current waveform of the LUT, the acquisition waveform of the camera, and the processed transient temperature waveform [17].

**Figure 16 sensors-24-02974-f016:**
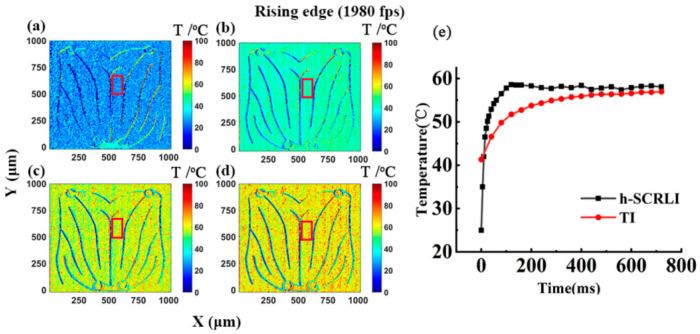
Transient two-dimensional temperature distribution of the rising edge of the blue LUT at 300 mA at (**a**) 0 ms, (**b**) 75 ms, (**c**) 95 ms, and (**d**) 125 ms, and (**e**) the comparison of the transient response of the h-SCRLI method and the thermal reflection imaging method [17].

**Figure 17 sensors-24-02974-f017:**
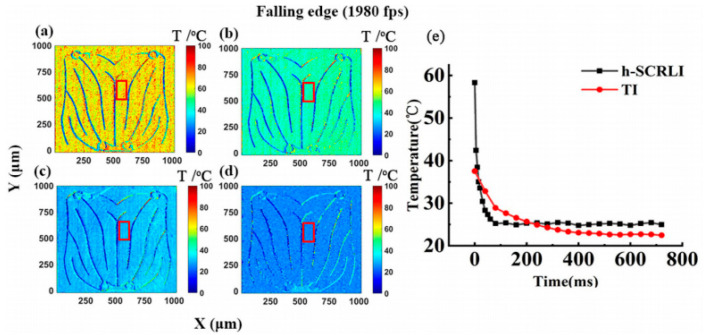
The evolving two-dimensional temperature profile of the blue LUT, propelled by a 300 mA current, is depicted at the descent of (**a**) 0 ms, (**b**) 5 ms, (**c**) 10 ms, and (**d**) 145 ms. Additionally, (**e**) presents a comparative analysis of the transient responses between the h-SCRLI method and the thermal reflection imaging method [17].

**Figure 18 sensors-24-02974-f018:**
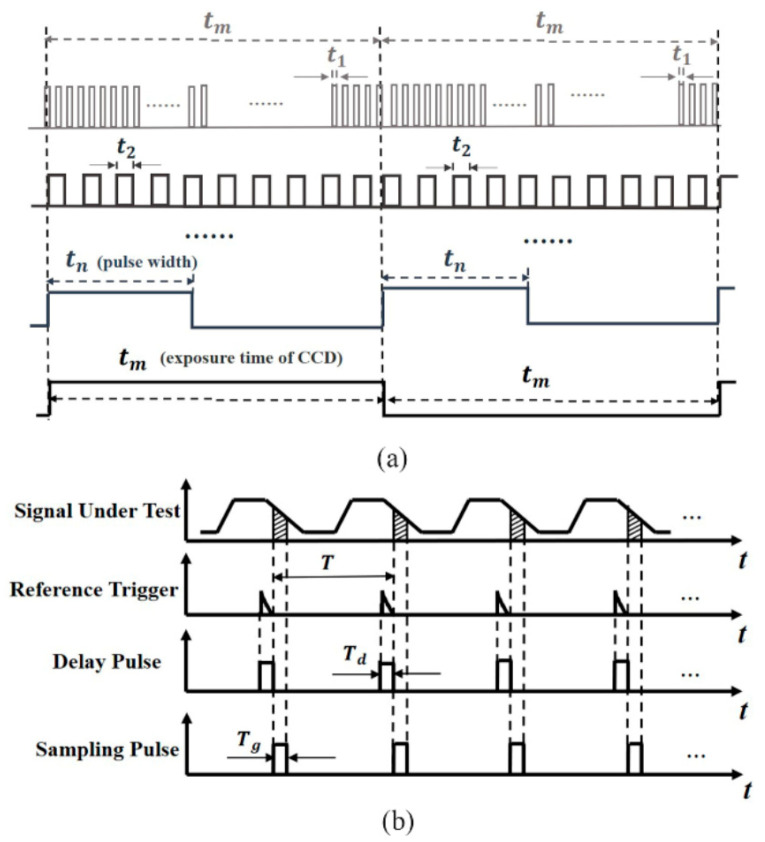
(**a**) The pulse signal diagram of driving the LUT (heating process); (**b**) Dropping sampling pulse setting (cooling process) [11].

**Figure 19 sensors-24-02974-f019:**
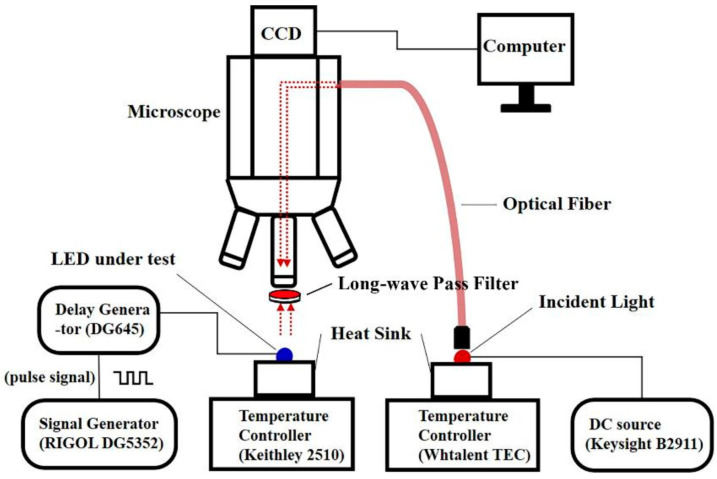
Schematic diagram of the experimental device [11].

**Figure 20 sensors-24-02974-f020:**
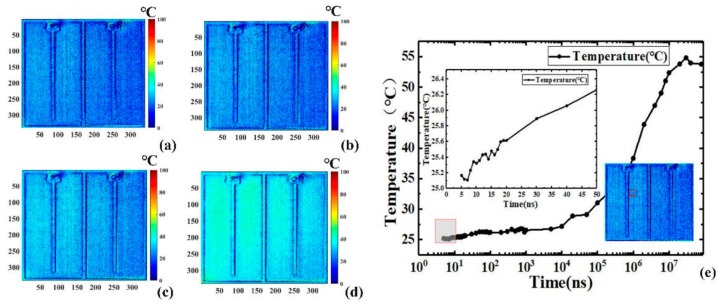
The transient two-dimensional temperature distribution of LUT driven by different pulse widths during the heating process of (**a**) 10 ns, (**b**) 500 ns, (**c**) 100 μs, and (**d**) 1 ms. (**e**) The time–response curve in the logarithmic coordinate of the average transient junction temperature in the red box [11].

**Figure 21 sensors-24-02974-f021:**
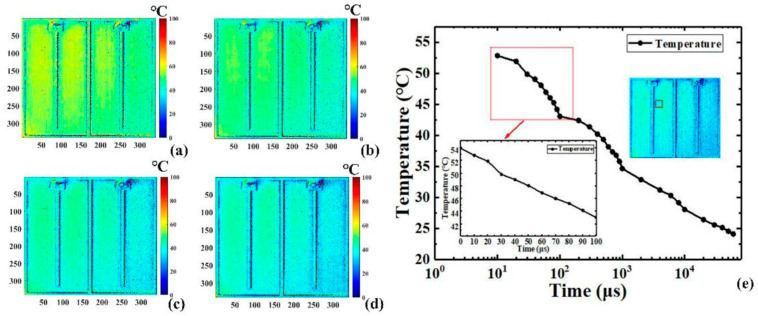
The transient two-dimensional temperature distribution of LUT driven by different pulse widths during the cooling process of (**a**) 0 μs, (**b**) 50 μs, (**c**) 100 μs, and (**d**) 200 μs. (**e**) The logarithmic coordinate time–response curve of the average transient junction temperature in the red box [11].

**Table 1 sensors-24-02974-t001:** Summary of the LED T_j_ steady-state measurement methods.

Measurement Method	Measurement Principle	Spatial Resolution	Advantages	Limitations
Micro-thermocouple	Seebeck effect	50 μm	Low cost	Limited resolution
			Readily available	Requires direct contact
Liquid Crystal Thermography (LCT)	Crystal phase transitions	2–5 μm	Low cost	Not a direct indicator of the T_j_
			good spatial resolution	Temperature resolution is limited to the liquid crystal
Infrared (IR) Thermography	Planck blackbody emission	3 μm	No contact	Typically limited spatial resolution
			Provides temperature maps	Environmental influences are prone to errors
TSOP	Spectral Power Distribution	2–3 μm	good spatial resolution	Expensive cost
			No contact	The relationship between temperature and optical parameters needs to be measured
Heterodyne Method	cycled phase lag	20 μm	No contact	Systematic errors cannot be corrected
			Simple to implement	Limited resolution

**Table 2 sensors-24-02974-t002:** Summary of the LED T_j_ transient measurement methods.

Measurement Method	Measurement Principle	Temporal Resolution	Advantages	Limitations
T3ster	Electrical	>100 ns	High current	Slow acquisition time
			PWM	Expensive cost
CRWM	Electrical	<200 ns	Error is less than FVM	One-dimensional drive measurements
			Fast acquisition time	Measures the average T_j_
Micro-Raman Spectroscopy	Phonon Frequency	>200 ns	Good spatial resolution	Slow acquisition time
			No contact	May require an unobstructed view of the device
Thermal reflection	Reflectivity	800 ps	No contact	Not direct indicator of the T_j_
			Good spatial resolution	May require an unobstructed view of the device
h-SCRLI	Reflectivity	Reach 68 μs at 14,600 fps	No contact	Not direct indicator of the T_j_
			Error is less than Thermal reflection	Acquisition time is determined by the camera
μ_HSTI	Reflectivity	5 ns	Fast acquisition time	Not direct indicator of the T_j_
			No contact	Difficult to detect changes in the temperature distribution.

## Data Availability

Not applicable.

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
