# Peer review of "LED Junction Temperature Measurement: From Steady State to Transient State"

_sensors, 2024, doi:10.3390/s24102974_

Round 1
Reviewer 1 Report
Comments and Suggestions for Authors
Authors report LEDs junction temperature measurement techniques by summarizing recent progresses. This review is clearly written, and I would recommend this manuscript for publication after authors solve the high match percentage (41%) of the iThenticate report.
Comments on the Quality of English Language
Minor editing of English language is required.
Reviewer 2 Report
Comments and Suggestions for Authors
The paper needs some major modifications.
1. The title is not appealing, it should be revised.
2. Please explain in the abstract part, why there is a need to write this review article and how’s this different from others.
3. The cited literature is too outdated. Please review the recent literature to catch the recent trends in the field.
4. Figures, 4, 8, and 9 are in poor image quality, please provide their high-resolution images.
5. Please add a table related to recent advancements in the field.
6. The incorporated equations are without references and not explained. Please fix this issue.
7. The authors should add future perspectives based on their study. An ideal review article should have this.
8. Extensive English corrections are required.
9. The summary is too generic, please modify it.
Comments on the Quality of English LanguageExtensive editing of English language required
Reviewer 3 Report
Comments and Suggestions for Authors
Reviewer's Comments
March 28, 2024
Sensors
Manuscript ID: sensors-2909486
Title: Advances in high-speed LED junction temperature measurement techniques
Comments:
1- Please update the references with recent ones (2023-2024) as the recent reference in your review article is 2022.
2- Please use the SI units especially with temperature.
3- Please add a flow chart after the introduction part to shows the techniques used for the measurement/detection of the LED junction temperature
4- Please add a table to summarize all techniques used for the measurement/detection of the LED junction temperature including advantages, disadvantages as well as the sensitivity and the detection limit.
After a clear thought, I highly recommend this manuscript for publication after careful consideration of the above mention comments,
Reviewer 4 Report
Comments and Suggestions for Authors
In this manuscript, the authors conduct an in-depth analysis and synthesis of LEDs junction temperature measurement techniques. This review explores hardware detection, experimental research methods, and testing approaches for high-speed signal junction temperature distribution in detail. Besides, the authors discuss the evolution from one-dimensional to two-dimensional LED junction temperature mapping and elucidates systematic construction involved in the experimental process as well as an alternative method to capture the junction temperature, addressing challenges posed by rapid transient conversion speeds beyond the camera's capability. This review has certain reference significance for researchers in this field. Therefore, I recommend this manuscript to be published in the journal of “Sensors” once the following problems have been solved.
(i) Many of the images in the manuscript are blurred, please replace them with high resolution original images. (Figure 4, 8b, 9)
(ii) In the reference part, there are some formatting errors (ref. 4, 5, 53).
Besides, the proportion of references after 2020 is too small.
Round 2
Reviewer 2 Report
Comments and Suggestions for Authors
Well revised.
Comments on the Quality of English LanguageModerate editing of English language required